# The Intestinal Barrier Dysfunction as Driving Factor of Inflammaging

**DOI:** 10.3390/nu14050949

**Published:** 2022-02-23

**Authors:** Eva Untersmayr, Annette Brandt, Larissa Koidl, Ina Bergheim

**Affiliations:** 1Institute of Pathophysiology and Allergy Research, Center for Pathophysiology, Infectiology and Immunology, Medical University of Vienna, 1090 Vienna, Austria; larissa.koidl@meduniwien.ac.at; 2Department of Nutritional Sciences, Molecular Nutritional Science, University of Vienna, 1090 Vienna, Austria; annette.brandt@univie.ac.at

**Keywords:** mucosal barrier, aging, inflammaging

## Abstract

The intestinal barrier, composed of the luminal microbiota, the mucus layer, and the physical barrier consisting of epithelial cells and immune cells, the latter residing underneath and within the epithelial cells, plays a special role in health and disease. While there is growing knowledge on the changes to the different layers associated with disease development, the barrier function also plays an important role during aging. Besides changes in the composition and function of cellular junctions, the entire gastrointestinal physiology contributes to essential age-related changes. This is also reflected by substantial differences in the microbial composition throughout the life span. Even though it remains difficult to define physiological age-related changes and to distinguish them from early signs of pathologies, studies in centenarians provide insights into the intestinal barrier features associated with longevity. The knowledge reviewed in this narrative review article might contribute to the definition of strategies to prevent the development of diseases in the elderly. Thus, targeted interventions to improve overall barrier function will be important disease prevention strategies for healthy aging in the future.

## 1. Introduction

In recent years, the function of the intestinal barrier has received increasing scientific attention as more and more intra- and extra-intestinal diseases, such as irritable bowel syndrome, inflammatory bowel diseases such as Crohn’s diseases, type 1 diabetes, colorectal cancer, acute inflammation-related diseases such as sepsis, and allergic diseases, were found to be associated with a dysfunctional intestinal barrier (for an overview, see [1,2,3]). The results of various animal studies demonstrated a link between intestinal barrier dysfunction and aging. For instance, aged monkeys had poorer intestinal barrier function, increased systemic inflammation, and higher microbial translocation compared to young animals [4,5]. In *Drosophila* models, intestinal barrier dysfunction has been shown to predict the approaching death of flies [6]. In this review, we want to explore whether intestinal barrier dysfunction and the accompanying alterations to the intestinal microbiota composition are driving factors for the increasing proinflammatory status during aging known as inflammaging.

Inflammaging was first described by Franceschi et al. in 2000 as a combination of a reduced ability to deal with stressors and the resulting increase in proinflammatory milieu (Figure 1) (for an overview, see [7]). More recently, inflammaging was defined as a “chronic, sterile, low-grade inflammation” that occurs during aging (for an overview, see [8]). A similar concept is metaflammation, describing a metabolically driven inflammation caused by nutrient excess (for an overview, see [9]).

## 2. Composition of the Intestinal Barrier in Health

The intestinal barrier is a highly complex structure composed of several layers that interact and influence each other (Figure 2).

Due to the major role of the intestinal barrier in preventing bacterial toxins and pathogens from the intestinal lumen entering into circulation, an impaired barrier function or even minor changes in the regulation of the epithelial, microbial, biochemical, or immunological barrier might contribute to aging-associated decline as well as disease development (see Section 3, Section 4 and Section 5). In the following Section, key components of this complex structure thought to be also critical in the aging-associated decline of intestinal barrier function are briefly described.

### 2.1. Intestinal Epithelial Layer

The gastrointestinal (GI) tract is the largest surface of the human body exposed to the external environment. Therefore, it not only fulfils functions such as food digestion, but also acts as a key line of defense, allowing for the selected survival of microbiota ingested but also prevalent in the GI tract and the entry of nutrients. Epithelial cells collaborate with immune and stromal cells to fight off pathogens, limiting their contact with the epithelium. Together with a stable microbiota, the mucus layer acts as first line of defense against external injuries [12,13]. Besides the pathogenic defense, the microbiota contributes to digestion of food and the production of vitamins, and promotes the development of the immune system [14,15]. Studies in germ-free mice demonstrated that intestinal microbiota affects the number of Peyer’s patches and lymphoid follicles and the general number of immune cells, e.g., IgA-producing plasma cells or CD8^+^ and CD4^+^ T cells [16].

The epithelial cell layer constitutes the core of the intestinal barrier. Cells found in this layer develop from pluripotent intestinal stem cells located in the crypts, which differentiate into a variety of cell types [17]. In human small intestine epithelial cells have a turnover of approximately 3.5 days [18]. The intestinal epithelial layer is composed of a multitude of different cell types, e.g., enterocytes, enteroendocrine cells, goblet cells, Paneth cells, or microfold cells (M cells) (Figure 2), all contributing to the complex interplay of nutrient absorption, while maintaining the mucosal barrier to avoid the permeation of bacterial toxins or pathogens and the secretion of immunological mediators (for an overview, see [19,20,21,22]).

Intestinal epithelial cells are tightly connected through junctional complexes composed of tight junctions on the apical side and adherence junctions and desmosomes towards the basolateral side (for an overview, see [23]). Accordingly, tight junctions are thought to be key components in the control of the paracellular transport in both the small and large intestine of the resulting semipermeable barrier (also see [22]). This semipermeable barrier facilitates the passage of ions and other substances, while the translocation of noxious molecules, such as bacterial toxins, is very limited [13,22,23,24]. Studies have shown that tight junctions are composed of transmembrane proteins, e.g., claudins, occludin, or junctional adhesion molecules, which interact with peripheral membrane proteins (e.g., Zonula occludens, ZO-1) (for an overview, see [24]). Results of in vitro studies also suggest that ZO-1 is linked to the cytoskeleton of the epithelial cell via F-actin [25]. It has been shown in vitro that the interaction between the actin–myosin cytoskeleton and the tight junction complex is critical in maintaining the paracellular barrier’s integrity, and this is mainly regulated by myosin light chain kinase (MLCK) [26]. These findings are supported by in vitro studies demonstrating that the activation of myosin light chains (MLC) by MLCK restructures the perijunctional F-actin and results in a reorganization of occludin and ZO-1, which increases permeability [27]. Moreover, the post-translational phosphorylations of occludin are critical for the opening and sealing of tight junctions, influencing the intestinal barrier function [28]. Depending on the site of phosphorylation, tyrosine phosphorylation attenuates the occludin–ZO-1 interaction, resulting in destabilized tight junctions, while the serine and threonine phosphorylation of occludin contributes to an intact tight junction assembly [29,30,31]. Furthermore, in vivo and in vitro studies recently demonstrated that post-translational nitration might result in a ubiquitin-dependent proteolytic degradation of tight junctions [32]. The results obtained in occludin knock-out mice suggest that while these mice suffer from histological abnormalities in various tissues, they exhibit morphologically intact tight junctions in intestinal tissue [33]. These studies suggests that tight junction complexes are complex units and that other mechanisms may also be critical in regulating intestinal barrier function. For instance, enteroendocrine cells secreting peptide hormones, such as ghrelin, peptide YY, cholecystokinin, and glucagon-like peptide 1, in response to nutrient exposure at the luminal side of the intestine (for an overview, see [34]) also secrete glucan-like peptide-2 [35]. The latter peptide has been shown to enhance intestinal barrier function through mechanisms involving the MLCK/pMLC signaling cascades [36,37]. Further studies are needed to fully elucidate the complex interplay underlying intestinal barrier function beyond tight junctions.

### 2.2. Biochemical Barrier and Immunological Barrier

The mucus layer coats the intestinal epithelial cells [12,13]. The main components are glycosylated mucin proteins, such as the mucin-2 being produced by goblet cells, [38,39]. Studies in mucin-2 knock-out mice demonstrate that these mice not only develop colitis, but also suffer from an impaired intestinal barrier function, underlining the role of mucus in maintaining intestinal homeostasis [40]. Besides the secreted mucins, there are also transmembrane mucins, which carry out barrier and signaling functions (for an overview, see [39]). The mucus layer is supported by the secreted antimicrobial peptides produced by secretory Paneth cells, contributing to the maintenance of the intestinal homeostasis (for an overview, see [41]). This is in line with studies in transgenic mice showing a reduced number of Paneth cells and the increased penetration of the intestinal barrier by commensal as well as pathogenic bacteria [42].

Bacteria-specific immunoglobulin (Ig) A, secreted by B-cells, further adds to a decreased penetration of bacteria and supports the mucosal barrier. The secretion of antigen-specific IgA is triggered by dendritic cells, which can be activated by epithelial M cells (located in follicle-associated lymphoid tissue as part of Peyer’s patches), contributing to antigen sampling and uptake from the intestinal lumen (for details please refer to [43,44]). A loss of IgA was suggested to be associated with an impaired intestinal barrier function in vivo [45]. However, M cells might also act as an entrance for pathogens (for an overview, see [44]).

## 3. The Aged Intestine: Alteration of Intestinal Barrier

### 3.1. Digestion and Absorption

The digestion and absorption of foods and drinks is a complex, multistage process depending on several endogenous factors, such as enzymes, GI motility, entero-endocrine hormonal activity, and the enteric nervous systems. Exogenous factors, such as food composition, but also microbial metabolite availability have been shown to affect digestion and absorption (for an overview, see [46,47]). Similar to many other physiological functions, the gastrointestinal function is also affected by degenerative processes and declines with aging. As also reviewed by others [48,49,50,51], olfactory, gustatory, and visual food perceptions, along with salivation and oral as well as dental health, decline in the third stage of life (for an overview, please also see [48]). Along with diminished appetite, which is associated with increased concentrations in appetite-related hormones, such as peptide-YY, leptin, and cholecystokinin, this is one of the key factors for malnutrition in older individuals above the age of 65 [52]. Motility and enzymatic and absorptive performances in the more distal part of the GI tract, e.g., the stomach and small and large intestines, have been suggested to be altered in the elderly, too (for an overview, see [48]). Some older studies reported an increase of 30–40% in gastric emptying time for solid and liquid foods in the elderly [53,54,55], which seemed also to be related to the presence of frailty [56]. The basal and stimulated secretion of pepsin, critical for protein digestion, was shown to decline with older age (~40% when comparing > 68–98-year-old humans with 18–34- and 35–64-year-old individuals) [57]. In contrast, in the small intestine, motility and transit time seem to be mainly unaffected by aging, while for the colon, reports are contradictory (for an overview, please see [48]). This might be related to the fact that colonic transit time can be affected by environmental factors, including physical activity and psychological and behavioral factors [58]. Furthermore, the digestion and absorption of nutrients in the elderly may also be affected by the quantity and activity of enzymes involved in the digestion of sugars, proteins, and fats such as saccharidases, proteases, and various lipases, which are shown to be altered in the elderly [59,60]. At least in rodents, the uptake of sugars and lipids seems to be altered, with the expression of fructose transporter Glut5 having been reported to increase, while glucose transporter Glut2 and lipid uptake and the expression of I-FABP have been shown to decrease [61,62,63]. Studies assessing sugar, amino acid, and lipid transporters in young and elderly humans are, to our knowledge, very limited or still missing. Holt et al. and Salles et al. reported in their reviews that lipid digestion and absorption in humans in general is well preserved in the elderly (for an overview, please see [64,65]). However, multi-drug intake, frequently observed in the elderly, might additionally influence the gastrointestinal barrier function and impact immune-mediated disease [66]. Still, drug intake is not only found in old-aged individuals, as recently reported in an Austrian population-wide study on anti-acid drug use [67].

### 3.2. Mucus and Mucosal Turnover

As mentioned above, the epithelium of the GI tract is a matter of constant and rapid renewal. Additionally, while cell proliferation and differentiation are affected in many organs of the human body, studies in elderly animals and humans suggest that the architecture of the intestinal epithelium in the small intestine is not markedly affected by aging [68,69]. It has been discussed that a hyperproliferative state balancing an increased rate in enterocyte apoptosis could be underlying this lack of morphological change in the small intestine (for an overview, see [48]). Accordingly, the total surface area available for digestive and absorptive processes in the small intestine seems not be markedly affected by aging. Studies in animals suggest an increased proliferation of colon mucosal cell proliferation along with decreased apoptosis [70].

In healthy older individuals, the thickness of the gastric and duodenal mucus layer is not altered [71]. In the ileum of old-aged mice, the number of goblet cells per villus displaying larger mucin granules may even be higher [72]. Tremblay et al. report a reduced expression of ileal α-defensins and lysozyme, two antimicrobial peptides produced by Paneth cells [72]. In old-aged mice, the thickness of the colonic mucus layer was reported to be reduced when compared to young animals, and this loss in elderly mice could be restored by supplementing *Lactobacillus plantarum* WCFS1 [73]. Furthermore, animal studies suggest that the effects of aging on the mucus layer in the GI tract may be affected by the sex of the host and that male mice may be more susceptible to aging-associated changes to the mucus layer than female animals [74]. In the same study, it was shown that, even in female mice with an ovariectomy, these sex-specific differences were still prevalent. Additionally, protein glycosylation is affected by aging. Together with the findings that the mucus layer may be critical in shaping the immunological properties of immune-competent cells, such as dendritic cells in the GI tract [75,76,77], it is discussed that changes in the glycosylation pattern of mucus in the GI tract might contribute to the alterations in the microbiota profile found in the elderly (please also see [78]). Further studies are needed to evaluate the impact of a changed mucus composition on aging-associated alterations to intestinal barrier function and microbiota composition.

### 3.3. Intestinal Immune System

Aging is associated with marked alterations of the immune response, also often referred to as immunosenescence [79]. While often used to describe the general decline in immune function in later stages of life, the word more correctly describes the changes associated with the aging of the immune system, to which senescent immune cells contribute (for an overview, also see [79]). In the following section, some of the key facts reported in recent years regarding aging-associated changes to the intestinal immune system are summarized. Recently, it was shown in a Japanese cohort of healthy individuals aged 35–81 years that the concentration of the Paneth cell α-defensin human defensin 5 (HD5) in feces was significantly lower in the elderly (age > 70 years) than in middle-aged individuals (age < 70 years) [80]. Paneth cell α-defensins have been suggested to be involved in the regulation of the intestinal microbiota composition [81]. Furthermore, antigen-specific T cell and B-cell responses that are critical in the protection against pathogens have been proposed to be affected by aging [82]. Booth et al. compared the T cell responsiveness of elderly people with younger individuals (>60 vs. <60 years of age) and reported that in tissue resident memory T cells (T_RM_) in ileum tissue, cellular characteristics, function, and number are affected by aging [83]. Furthermore, as reviewed by Galletti et al. [84], in aging mice, the induction of tolerance as well as the humoral response to the oral administration of ovalbumin are decreased. Our own data suggested that oral immunizations induced both allergen-specific IgG1 and IgG2a in aged animals, while adult animals preferentially developed IgG1. Of interest, when gastric digestion was impaired, both age groups developed comparable levels of allergen-specific IgE upon oral immunizations [85]. The alterations of oral tolerance and intestinal immune response might be related to a decrease in dendritic cells, as well as changes in the architecture of Peyer´s patches and a dysregulation of T cells. Kato et al. reported that in aged mice, feeding them OVA produced selected T(h)2- but no T(h)1-type cytokines in CD4(+) T cells from Peyer’s patches [86]. Moreover, the ability of mucosal dendritic cells in small intestine of old mice to stimulate TGFβ secretion and differentiate CD4(+) LAP(+) T cells was also reported to decline [87]. In the same study, the frequencies of regulatory-type IEL subsets, such as TCRγδ(+) and TCRαβ(+)CD8αα(+), were shown to be lower in the gut mucosa of aged mice. Kobayashi et al. reported that in old-aged mice (18 months of age), M-cell density in the follicle-associated epithelium is reduced and that, subsequently, the ability to transcytose particle luminal antigens is decreased [88]. The results of our own studies show that aging in mice is associated with a decrease in F4/80 positive cells and NOx in the small intestine, suggesting that the number of macrophages, and maybe also the production of reactive nitrite species, may be altered [89]. Changes in the macrophage responses in mice in relation to aging-associated changes of intestinal barrier function were confirmed by others [90].

Not all components of the intestinal immune system are altered by aging, and further studies are warranted to assess how the resulting changes in the immunological interplay in the intestinal mucosa affect intestinal microbiota composition and barrier function, as well as the different gut–peripheral tissue axis.

### 3.4. Intestinal Barrier and Junction Proteins

While the gross architecture of the small intestinal epithelium seems rather unchanged, even at older age, studies in humans and animals assessing intestinal permeability and tight junction proteins suggest that aging is associated with a loss of tight junction proteins in the small and large intestine. Already in the 1980s, Hollander et al. reported that older age in rats is associated with an increased permeability to larger macromolecules [91]. Studies in non-human primates, such as aged baboons, reported a reduced expression of the tight junction proteins ZO-1, occludin, and junctional adhesion molecule-A (JAMA), leading to an increased permeability for horseradish peroxidase [92]. In contrast, in the human ileum, the expression of ZO-1, occludin, and JAMA-1 mRNA and protein was not altered in older individuals (aged 67–77 years) when compared to younger controls (20–40 or 7–12 years). Moreover, the permeability to macromolecules was not different between age groups [93]. However, in this study, intestinal permeability was assessed using an ex vivo approach. As mentioned above, tight junction proteins, such as occludin, are regulated through phosphorylation [28,29,30,31]. Data on the effects of aging on phosphorylation of tight junction proteins are still scarce. Interestingly, changes to intestinal integrity with aging have also been proposed in non-vertebrates, including *Drosophila* and *C. elegans* (for an overview, see [94,95,96]).

This loss of tight junction proteins and the increased permeability of macromolecules in rodents [91,97,98] was shown to be associated with elevated bacterial toxin levels, especially bacterial endotoxin in diseases of various etiologies (for an overview, see [12,99]). The results of our own group suggest that the aging-associated changes in intestinal microbiota composition are also associated with increased bacterial endotoxin levels in the portal and peripheral blood [89,100] and with an increased expression of Toll-like receptors in the liver (see Section 5). Kühn et al. prevented the loss of tight junctions and increased intestinal permeability by altering the availability of intestinal alkaline phosphatase, which may not only result in less frailty, but also extend lifespan [98]. The results of several in vivo and in vitro studies suggest that in the aging-related loss of intestinal barrier function, changes in stem cell proliferation and the alterations of immune cells and the composition of mucins, as well as changes in intestinal microbiota, may be critical factors [78,88]. Other alterations of intestinal stem cells, including chronic activation and misdifferentation, may also be critical in the development of intestinal barrier dysfunction in aging [78,101]. Herein, the excessive formation of reactive oxygen species through both extrinsic and intrinsic measures, including a “Warburg-like reprogramming” [102], seem to be relevant, as shown by results obtained in studies using *Drosophila*. A changed mucus layer, e.g., changes in the glycosylation pattern of mucins [78], in anti-microbial peptides [72], and in levels of soluble immunoglobulin A [103,104,105], as well as the prevalence of immune cells, e.g., M cells [88] and T cells [106], have also been associated with aging-associated intestinal barrier dysfunction (for an overview, see [78]). Results of in vitro studies in Caco-2 cells suggest that TNFα may alter MLCK protein levels, which are involved in the regulation of tight junction proteins in the small intestinal epithelium [107,108]. Interestingly in our own studies, older age in mice was associated with lower NO_x_ and F4/80 mRNA expression, suggesting that TNFα may not predominantly stem from macrophages in the small intestinal tissue [89]. However, to date, the molecular mechanisms underlying the loss of tight junction proteins and, subsequently, the increased permeability and translocation of bacterial toxins in elderly has not been clarified.

## 4. The Microbiome in the Aging Intestine

From lifespan experiments in animals [109,110] to the analysis of the microbiome over the human lifespan (for an overview, see [111,112]) and of centenarians [113,114], a plethora of evidence shows that the microbiome is not only associated with diseases, e.g., gastrointestinal diseases such as IBS (for an overview, see [115]) and IBD (for an overview, see [116]), diabetes, metabolic liver disease [117], and allergies (for an overview, see [118]), but also aging (for an overview, see [119]) and inflammaging (for an overview, see [8]).

When evaluating the microbiome, two factors are commonly considered that have a substantial impact on health and disease: stability and diversity, with a more stable and diverse microbiome generally being associated with better health (for an overview, see [120]). Starting with birth [121], the individual microbiome is subject to change throughout life [111,122]. The greatest alterations occur during our infancy when the personal microbiome is in its developing stage. This stage is followed by the adult microbiome, which has the highest stability. With further aging, the microbiome becomes again less stable (Figure 3) [123] (and for an overview, see [111,124,125]).

Difficulties arise when comparing the taxonomical diversity of microbiota between individuals. The microbiota composition is highly susceptible to exogenous and endogenous factors, meaning that making comparisons between different study populations is a difficult task. It is currently impossible to make universally valid statements about the average healthy microbiome. While the taxonomical composition is highly diverse, the metabolic pathways of the microbiome of healthy individuals are much more comparable [127]. As several bacterial species are able to perform the same biological function, the functional core hypothesis suggests that the bacterial species themselves are interchangeable, as long as the gene for the core metabolic pathways are present (for an overview, see [120,128]). Not surprisingly, the developing microbiome of children shows not only compositional, but also functional differences compared to adults [129]. This might also be applicable to the elderly. Several studies have shown that the microbiome of centenarians differs in composition and metabolic pathways compared to other age groups. Biagi et al. reported a generally less diverse microbiome in centenarians compared to young and non-centenarian elderly. Herein, especially, facultative anaerobes from the phylum Proteobacteria, such as species from *Escherichia*, *Haemophilus*, *Klebsiella*, *Proteus*, *Pseudomonas*, etc., and additionally *Bacillus* and *Staphylococcus*, were found to be increased in fecal samples. *Eubacterium limosum* was reported to be increased 15-fold in centenarians, and the authors suggested that *E. limosum* and relative species might be characteristic for centenarians [113]. Kim et al. also reported differences in the microbiome compositions of centenarians compared to younger age groups of the same region [130]. They observed differences in the average phyla composition, with centenarians (aged 95 to 108 years) having a greater phyla diversity than younger age groups (aged younger than 80 years). Compared to non-centenarian elderly (aged 67 to 79 years), centenarians showed, on a phylum level, a higher relative abundance of Verrucomicrobia and, on a genus level, a higher relative abundance of *Akkermansia*, *Clostridium*, *Collinsella*, *Escherichia*, and *Streptococcus*, but a lower abundance of *Faecalibacterium* and *Prevotella*. Compared to adults (aged 26 to 43 years), centenarians showed a higher relative abundance of Proteobacteria and Actinobacteria, in addition to Verrucomicrobia. The abundance of Proteobacteria was also higher in non-centenarian elderly than in adults, suggesting an increase in the share of Proteobacteria with age in this study population. Results from a study comparing *Bifidobacterium* strains isolated from the feces of healthy young-to-middle-aged adults (30–40 years old) and elderly (>70 years of age) revealed that the adhesive ability of the human *Bifidobacterium* microbiota was reduced with the increasing age of the host [131]. Centenarians had more microbiota with pathways in the phosphatidylinositol signaling system, glycosphingolipid, and N-glycan biosynthesis [130]. A 2021 study observed that the microbiome of Japanese centenarians is not only enriched by species from *Alistipes*, *Parabacteroides*, *Bacteroides*, *Clostridium*, and *Methanobrevibacter* compared to young and elderly controls, but also by bacterial species in possession of genes with certain rare metabolic pathways in the bile acid metabolism (e.g., *Clostridium scindens*) [114]. Compared to young controls and non-centenarian elderly, levels of primary bile acids were decreased, but those of their metabolites (secondary bile acids) increased, which is unique to centenarians. The level of the bile acid isoalloLCA was increased, which was reported to show potent antimicrobial activity against Gram-positive bacteria. The study also observed that short-chain fatty acid (SCFA) levels were decreased and branched SCFA and ammonium levels were increased. The authors suggested that this might be caused by a reduced abundance of SCFA-producing strains, while the amino acid metabolizers increased, likely due to a reduced upper intestinal proteolytic capacity in the elderly. In contrast, old mice were reported to have increased ratios of primary to secondary bile acids in the liver, serum, and intestines, but the ratio was reduced by microbiota remodeling via co-housing with younger animals [132]. Under regular conditions, more than 90% of the bile acids (primary bile acids) are synthesized in the hepatocytes, while under 10% (secondary bile acids) are produced in the gut through metabolization by the microbiota. Secondary bile acids have also been reported to have strong antimicrobial activity [114,133] (and for an overview, see [133]) and are able to inhibit bacterial overgrowth. Bile acids have several roles besides the support of dietary lipid absorption, as they are able to activate pathways regulating the metabolism of bile acids, lipids, and carbohydrates and inflammatory processes (and energy expenditure) via the receptors farnesoid X receptor and G protein-coupled membrane receptor 5 [133]. Bile acid homeostasis, therefore, plays an important role in the maintenance of physiological conditions. It has further been suggested that bile acid composition and metabolism, as well as microbiota-associated digestive capacities, decrease with increasing age in humans [134,135,136].

In animal studies, it could be observed that inflammaging is associated with the gut microbiome. Compared to young mice, older animals are reported to have higher levels of inflammatory cytokines, such as systemic IL-6 [90,137] and TNFα [90,138], and higher fecal lipocalin-2 levels, indicating intestinal inflammation [137]. Older mice also had a greater diversity than young animals, and a significant difference in composition between old and young mice was observed [137]. An elegant mouse study by Thevaranjan et al. was able to link inflammaging to the microbiome [90]: old mice were observed to have higher IL6 and TNFα levels not only compared to young controls, but also compared to germ-free animals. Of interest, the anti-bacterial capability of the macrophages was reduced in old wild type mice compared to young, germ-free and old TNFα knock-out mice. The authors suggested that the higher TNFα levels in old mice results in the impairment of the macrophages and is caused by the microbiota, as the macrophages of aged TNFα knock-out mice or germ-free mice were not impaired in their anti-bacterial activity. Furthermore, they also observed that intestinal permeability increased with mouse age. The author suggested that inflammaging is dependent on the microbiota. By colonizing young and old germ-free mice from young and old donors, microbiota composition and an aged microbiome contribute to inflammaging. Thevaranjan et al. also reported that a higher proportion of germ-free mice lived to the age of 600 days than their conventional counterparts. Results of the same study also suggest that aging-associated intestinal microbiota dysbiosis and the increase in TNFα found in old-aged mice may be involved in the increased intestinal permeability. The transferal of an aged microbiome to young and old germ-free mice indicates that the aged microbiome can induce increased levels of several Th cell subsets in the spleen, inflammation in the small intestine, and characteristics of immunosenescence. TNF-α levels were increased in groups with the aged microbiome [138]. Furthermore, aging-associated changes in the intestinal microbiota shown to accompany aging in several species, including humans [89,139,140,141,142], have been linked to a chronic activation of the JAK/Stat signaling cascades. It has been suggested by studies in model organisms that the inhibition of these pathways may prevent not only dysbiosis, but also age-related metaplasia [140].

## 5. Consequences of Aging-Associated Alteration of Intestinal Microbiota and Barrier Function: The Liver and the Brain as Examples

The alterations found at the level of intestinal microbiota composition and intestinal barrier function in aging have been proposed to be interlinked with aging-associated decline in other organs. In the following section, some of the key findings on the interaction of aging-associated alterations at the level of the gut and the liver, the latter being also often referred to as the “metabolic control center”, as well as the brain as the key regulator of cognition and superordinated center of physiological control, are summarized.

### 5.1. Liver

The results of several epidemiological studies suggest that aging is an independent risk factor for the development of liver-related diseases [143]. Indeed, it has been suggested that impairments of hepatic function, e.g., the ability to clear substances such as bacterial toxins that enter the organism through the GI tract, may contribute to the age-related increased susceptibility to disease development of many other organs [144]. Older age (>65 years) by itself may be associated with elevated liver enzymes, even in the absence of any overt disease [145]. The results of studies in various species, including humans, suggest that a loss of liver volume, decreased hepatic blood flow, and morphological changes, including changes of volume of hepatocytes and the accumulation of dense bodies, are the results of aging [143,146,147]. Data on aging-associated alterations of liver structure and function are still contradictory. Ogrodnik et al. [148] suggested that old age in mice is associated with the development of hepatic steatosis. In our own studies, we found very limited to no accumulation of fat in the liver; rather, old age was associated with the development of inflammation and fibrosis in the liver [89,100]. Despite a continuously increasing number of studies, it has not yet been fully clarified whether or not aging intrinsically compromises the liver in humans and animal models.

Due to its anatomical location receiving a more or less ’unfiltered’ blood from the gut, the liver is confronted not only with nutrients, but also many xenobiotics, as well as bacterial toxins and metabolites stemming from the intestinal microbiota, along with endocrine mediators. This allows for a rather direct communication between the gut and the liver. However, the so-called gut–liver axis is a bi-directional exchange pathway (for an overview, also see [149]). The liver communicates with the gut through the synthesis of bile acids released in the intestine via the biliary tact. Changes to the intestinal microbiota composition, but even more so an increased permeability of the intestinal barrier, have been shown to be associated with aging (see Section 3, Section 4 and Section 5). This leads to changes in the exposure of the liver towards bacteria-derived metabolites and toxins, which in turn can lead or add to the activation of signaling cascades, and herein especially the Toll-like receptor-dependent signaling cascades in the liver. The latter have been proposed to add to the pro-inflammatory state found in older age. Indeed, we recently showed that not only is the expression of toll-like receptor 4 (TLR4) induced in older-aged mice, but so too are other TLRs in the liver [89,100]. Furthermore, the genetic deletion of the lipopolysaccharide binding protein (LBP), thought to be required for the recognition of bacterial endotoxin [150], was associated with a damping of aging-associated changes in the liver tissue. The studies of Chung et al. [151] further suggest that the livers of old-aged rats are more susceptible to LPS, with a more pronounced LPS-dependent upregulation of pro-inflammatory IL-1β/inflammasome pathways in the liver. Furthermore, it also has been shown that a supplementation of intestinal alkaline phosphatase found in the intestinal brush border dampens aging-associated increases in liver enzymes [98]. Further studies are needed to fully understand the interplay of the gut and liver in aging-associated liver decline.

### 5.2. Brain

During aging, not only are the liver and other organ systems affected by aging-related changes. For instance, cognitive aging is also part of the normal aging process and is accompanied by, e.g., slowed down reaction process or mild short-term memory loss, while, e.g., knowledge and acquired skills seem to persist [152,153]. Aging is an independent risk factor for neurodegenerative disease [154]. Almost 16.5% of the European population aged ≥65 years suffers from cognitive impairments [155], while a recently published systematic review on the global prevalence of cognitive impairments resulted in values ranging from 5.1 to 41% in people older than 50 [156]. Nearly ~7.1% of the European population suffer from dementia [157], for which cognitive impairments count as a kind of transitional state, and which is, e.g., estimated in the US to account for 13.6% of deaths [158]. However, the cause and molecular mechanisms of age-associated cognitive impairments are complex and still not fully determined. Aging-associated cognitive decline is described as a multifactorial process in which mitochondrial dysfunction, the accumulation of dysfunctional and aggregated molecules, an impaired lysosome and proteasome function, and neuronal calcium homeostasis, as well as impaired DNA repair, contribute to cellular and molecular mechanisms in brain aging (for an overview, see [159]). In recent years, alterations of the so-called “gut–brain–axis” (for an overview, see [160]) increasingly receive attention and are discussed to be involved in aging-associated cognitive impairments.

Lee et al. demonstrated that gnotobiotic mice receiving an aged microbiome suffered from depressive-like behavior and impaired short-term and spatial memory [161]. Compared to younger controls, it has been shown that 15-month-old C57BL/6 mice suffered from cognitive impairments, which were accompanied by altered microbiota composition and an impaired intestinal barrier function [162]. This is also in line with findings of studies in senescence-accelerated mouse models [163,164]. Wu et al. further showed that the impaired intestinal barrier function in old-aged mice is associated with a loss of tight junction proteins in the small intestine and an increased concentration of bacterial endotoxins in the blood. In this study, TLR4-dependent signaling cascades were induced in both the intestine as well as brain tissue [162]. Spatial memory was enhanced in old-aged TLR4 KO mice compared to wild-type mice of a similar age in a sex-dependent manner [165]. The systemic inflammation associated with the increased translocation of bacterial (endo)toxins (for an overview, see [166]) has been discussed to further contribute to cognitive impairment during the aging process in humans [167]. Indeed, as demonstrated in the study of Lin et al., older, healthy adults (~71.2 years) suffer more from increased inflammatory markers (e.g., IL-6) associated with cognitive age-related deficits than younger (~22.3 years) participants [167]. This hypothesis is further supported by studies performed in old mice suffering from memory defects and impaired cognition due to acute inflammations, as well as in mouse models of neurodegeneration [168,169]. Old patients with prodromal Alzheimer’s disease suffer from increased LPS levels [170] and altered microbiota compositions [171,172]. Targeting the intestinal microbiota composition with probiotics or single bacterial species, e.g., *Akkermansia muciniphila*, in mice with cognitive deficits has been found to protect the intestinal barrier and improve cognitive functions [164,173]. This has been shown not only with mild cognitive impairments, but also in several studies in mouse models of Alzheimer´s disease (AD) (the most common form of dementia), as well as in patients with AD reported to have altered microbiota composition [163,174,175,176,177,178], which is associated with decreased diversity and altered composition [176]. So far, no single strain of bacteria has been identified to be crucial during the development of cognitive impairments [179]. While only being a case report, it was shown in a 90-year-old patient with AD that a fecal microbiome transfer (FMT) used to treat a *Clostridioides difficile* infection was associated with an improved cognitive function test [180]. While the modulation of the gut microbiota via FMT has gained increasing interest for the treatment of *Clostridioides difficile* infection, inflammatory bowel diseases, obesity, metabolic disease, and, as was lately even suggested, cancer therapy (for an overview, see [116,181,182]), its application in age-related diseases still needs further research. First clinical trials in old AD patients treated with a probiotic milk (*Lactobacillus acidophilus*, *Lactobacillus casei*, *Bifidobacterium bifidum*, and *Lactobacillus fermentum*) for 12 weeks also demonstrated positive effects on cognitive function, assessed by a mini-mental state examination [183]. However, as discussed in a recently published meta-analysis of randomized controlled trials, the study situation on the effects of probiotics in patients with dementia is not yet clear [184].

Beside the pathway mediated through an activation of the immune system, further mechanisms are thought to link the microbiome to the central nervous system. As reviewed by Sampson et al., a direct activation of the vagus nerve or alterations of the production and release of neurotransmitters, hormones, or other metabolites (e.g., serotonin or SCFA) was able to cross the blood–brain barrier (for an overview, see [185]). This by no means precludes that, similarly to the intestinal barrier, the blood–brain barrier by itself is altered during aging [186], contributing to the increased LPS levels found in brains of old-aged mice [162]. Further studies are necessary to delineate the molecular mechanisms affecting the gut–brain axis and, subsequently, their contribution to cognitive dysfunction during aging.

## 6. Future Perspectives

As summarized in this review, the gastrointestinal epithelial barrier, with its multiple layers and its various function, is affected by the physiological aging process. However, as its role in healthy aging or disease development becomes increasingly evident, attempts to restore the barrier function, e.g., through modulation via microbiota modifications, supplementing strains such as *Lactobacillus plantarum* [187] or their metabolites [188,189], are of special interest. Besides microbiota modulation through probiotic strains or postbiotics, the current and future treatment of epithelial barrier dysfunction could include nutritional interventions and also bioactive pharmaceutical molecules, biologicals, or mucoprotectants (for an overview, see [190]). Recently, the disease-preventive effect of the zonulin antagonist larazotide acetate was demonstrated for the prevention of arthritis [191]. In the future, targeted epigenetic and miRNA approaches might also be employed in the restoration of disrupted epithelial barriers (for an overview, see [190]). We are far from understanding the detailed pathways of intestinal barrier dysfunction contributing to aging. With ongoing demographic changes, healthy aging supported by optimal mucosal barrier functionality will become increasingly important and needs the specific focus of research efforts in the future.

## Figures and Tables

**Figure 1 nutrients-14-00949-f001:**
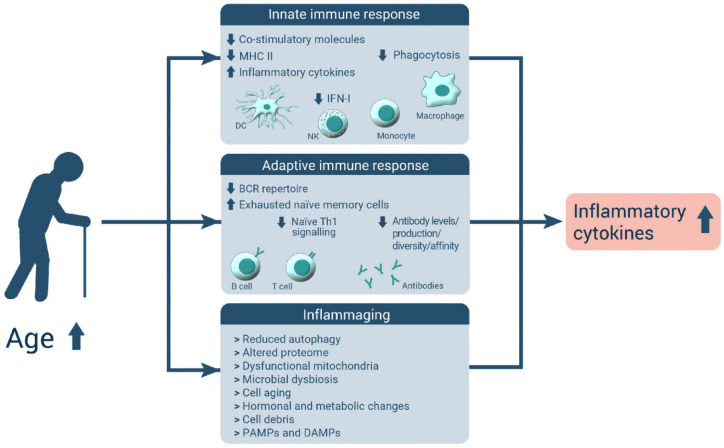
Concept of inflammaging. Due to immunosenescence associated with changes in the innate and adaptive immune response, inflammatory pathways are increasingly activated, leading to a constant low-grade inflammatory state also termed inflammaging (modified after [10]). Abbreviations used: Up-arrow–increased; down-arrow—reduced; MHCII—major histocompatibility complex II; IFN-I—type-I interferon; DC—dendritic cell; NK—natural killer cell; BCR—B-cell receptor; Th1—type 1 T helper cell; PAMPs—pathogen-associated molecular patterns; DAMPs—damage-associated molecular patterns.

**Figure 2 nutrients-14-00949-f002:**
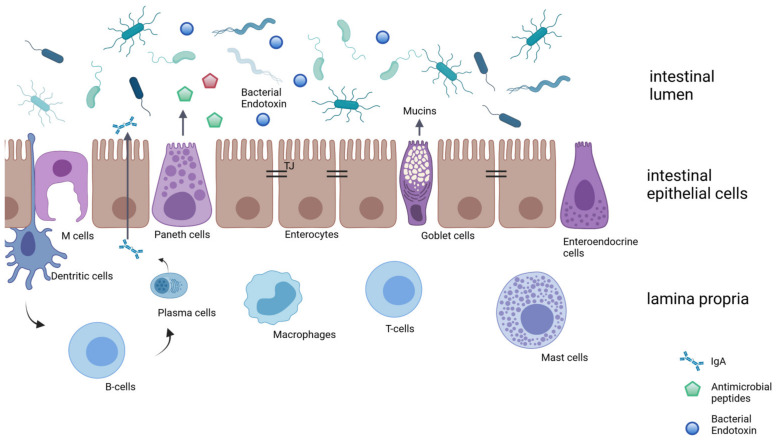
Schematic drawing of the intestinal barrier. The intestinal barrier includes a microbial, biochemical, physical, and immunological barrier. While the biochemical barrier consists of mucins and antimicrobial peptides, the physical barrier is composed of the epithelial monolayer with, e.g., enterocytes, goblet cells, enteroendocrine cells, Paneth cells, and microfold cells (M cells). Within the lamina propria, different immunological cells interact with the intestinal barrier, e.g., macrophages, dendritic cells, T cells, or mast cells. IgA: Immunoglobulin A. Modified after [3,11,12] and created with BioRender.com (accessed on 16 February 2022).

**Figure 3 nutrients-14-00949-f003:**
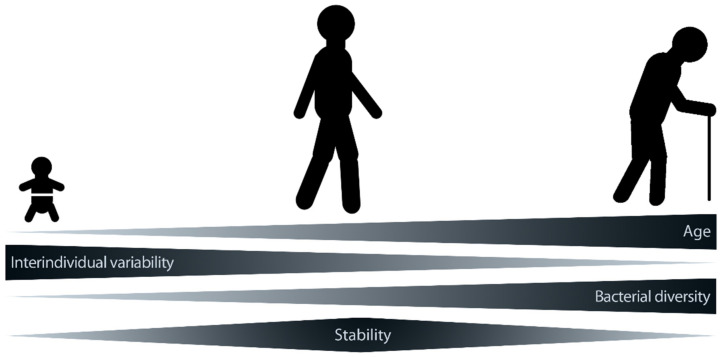
Stability of microbiota composition over the lifetime. During one’s lifetime, not only does the microbial composition change, but so too does its stability and diversity, which contributes to a more resilient microbial composition in adults (modified after [126]).

## Data Availability

Data sharing not applicable. No new data were created or analyzed in this study. Data sharing is not applicable to this article.

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
