# Peer review of "The Intestinal Barrier Dysfunction as Driving Factor of Inflammaging"

_nutrients, 2022, doi:10.3390/nu14050949_

Round 1

Reviewer 1 Report

In the manuscript titled "The intestinal barrier dysfunction as driving factor of inflammaging " Eva Untersmayr and colleagues, have reported that even though it remains difficult to define physiological age-related changes and distinguish them from early signs of pathologies, studies in centenarians provide insight into intestinal barrier features associated with longevity. Knowledge reviewed in this narrative review article might contribute to the definition of strategies to prevent the development of diseases in the elderly. Thus, targeted interventions to improve overall barrier function will be important disease prevention strategies for healthy aging in the future. I have a few comments regarding the present manuscript.

-The manuscript in general is a good piece of work with important information, maybe more detailed information about specific microbial signatures is required.

-Which specific treatments could improve the intestinal barrier, maybe this could add more relevance to the present manuscript

-In some cases, some typos are found in the name of different strains

-Please prefer the word microbiota instead of flora

-In section 4, the lifespan, the authors have mentioned animal studies, more detailed information about human studies is required here

-The authors have introduced in the final lines FMT, maybe more information about this is required

-An idea, the authors have state some lines to future actions, maybe that is required in a new section "Future perspectives"

Author Response

In the manuscript titled "The intestinal barrier dysfunction as driving factor of inflammaging" Eva Untersmayr and colleagues, have reported that even though it remains difficult to define physiological age-related changes and distinguish them from early signs of pathologies, studies in centenarians provide insight into intestinal barrier features associated with longevity. Knowledge reviewed in this narrative review article might contribute to the definition of strategies to prevent the development of diseases in the elderly. Thus, targeted interventions to improve overall barrier function will be important disease prevention strategies for healthy aging in the future. I have a few comments regarding the present manuscript.

We appreciate your feedback and your time and expertise summarized in your comments. The comments have been addressed in the amended version of the manuscript. We believe that implementing the suggested changes, improved our manuscript.

-The manuscript in general is a good piece of work with important information, maybe more detailed information about specific microbial signatures is required.

We thank the reviewer for the positive feedback. We have included more detailed information on specific microbial signatures in the revised version of the manuscript (lines 332, 337-346, 353-354).

-Which specific treatments could improve the intestinal barrier, maybe this could add more relevance to the present manuscript.

We thank the reviewer for this important comment. In the amended version of the manuscript, we have included information on current and future therapeutic approaches for epithelial barrier dysfunction in the chapter “Future perspectives” (lines 530-536). Moreover, we have added 2 references (Ref. 190 and 191).

-In some cases, some typos are found in the name of different strains

Thank you very much. All genera and species names were changed to italic. In line 35 typo was corrected.

-Please prefer the word microbiota instead of flora.

This was changed as suggested.

-In section 4, the lifespan, the authors have mentioned animal studies, more detailed information about human studies is required here.

We fully agree and have included 2 additional references to highlight knowledge from human studies (Ref. 111, 112).

-The authors have introduced in the final lines FMT, maybe more information about this is required

We have included more information as suggested and have added 3 new references (Ref. 116, 181 and 182).

-An idea, the authors have state some lines to future actions, maybe that is required in a new section "Future perspectives".

Thank you for this excellent suggestion. We have changed the last section and have added new information on treatment of barrier dysfunction (see also comment above).

Reviewer 2 Report

In this manuscript, the authors review the field of inflammaging and the role of the intestinal barrier as a contributing or causative factor. Overall, this is a comprehensive review covering a relatively new topic. While the English is mostly correct, a quick review of grammar would be needed  (see minor comments below). 

Minor edits: 

-L12: comma after "barrier"

-L35: Drosophila should be italicized

-L36 spacing is off

-Figure 2: no space between "plasmacells" of "Mastcells" in the figure

-L82: do not abbreviate approximately unless that is a specific abbreviation of the journal.

-Minor suggestion that the paragraph beginning at L79 could be broken into 2 paragraphs with the part about tight junctions becoming a 2nd paragraph.

-L124: comma after "function"

-L128: comma after "Paneth cells"

-L130: can delete "which suffer from"

-L138: switch words from "act also" to "also act"

-L169: delete "alike"

-L173: spelling error (durg) and comma missing after "intake"

-L216: should read "Paneth cell alpha-defensins" instead of "Paneth cells alpha-defensin"

-L244: "warrant" should be "warranted"

-L251: please change "80s" to "1980s" 

-L263: italicize "Drosophila"

-L279-280: the meaning is currently unclear. please fix this sentence

-L325: is this the phylum Proteobacteria and the class Bacilli? Just clarifying because those are not usually combined in this way.

-L392: do you really mean "chapter" or is it technically a section or subsection?

-L414: comma after "confronted" and after "nutrients"

-L437: sentence is a little unclear. please simplify.

-L504: "ageing" should be "aging"

Author Response

In this manuscript, the authors review the field of inflammaging and the role of the intestinal barrier as a contributing or causative factor. Overall, this is a comprehensive review covering a relatively new topic. While the English is mostly correct, a quick review of grammar would be needed  (see minor comments below). 

Thank you very much for your positive review and acknowledgement of our work. Your comments have been addressed in this revised version of the manuscript. We believe that implementing the suggested changes, improved our manuscript.

Minor edits: 

-L12: comma after "barrier"

-L35: Drosophila should be italicized

-L36 spacing is off

-Figure 2: no space between "plasmacells" of "Mastcells" in the figure

-L82: do not abbreviate approximately unless that is a specific abbreviation of the journal.

-Minor suggestion that the paragraph beginning at L79 could be broken into 2 paragraphs with the part about tight junctions becoming a 2nd paragraph.

-L124: comma after "function"

-L128: comma after "Paneth cells"

-L130: can delete "which suffer from"

-L138: switch words from "act also" to "also act"

-L169: delete "alike"

-L173: spelling error (durg) and comma missing after "intake"

-L216: should read "Paneth cell alpha-defensins" instead of "Paneth cells alpha-defensin"

-L244: "warrant" should be "warranted"

-L251: please change "80s" to "1980s" 

-L263: italicize "Drosophila"

All edits have been implemented in the revised version of the manuscript.

-L279-280: the meaning is currently unclear. please fix this sentence

The sentence has been adapted.

-L325: is this the phylum Proteobacteria and the class Bacilli? Just clarifying because those are not usually combined in this way.

Thank you very much for pointing this out. This was clarified in the text (lines 331-333).

-L392: do you really mean "chapter" or is it technically a section or subsection?

-L414: comma after "confronted" and after "nutrients"

These edits have been implemented in the revised version of the manuscript.

-L437: sentence is a little unclear. please simplify.

The sentence has been adapted.

-L504: "ageing" should be "aging"

Thank you very much. This has been changed throughout the manuscript.